# Impact of Three Kinds of Early Interventions on Developmental Profile in Toddlers with Autism Spectrum Disorder

**DOI:** 10.3390/jcm11185424

**Published:** 2022-09-15

**Authors:** Francesca Cucinotta, Luigi Vetri, Liliana Ruta, Laura Turriziani, Loredana Benedetto, Massimo Ingrassia, Roberta Maggio, Eva Germanò, Ausilia Alquino, Rosamaria Siracusano, Michele Roccella, Antonella Gagliano

**Affiliations:** 1I.R.C.C.S. Centro Neurolesi Bonino Pulejo, 98124 Messina, Italy; 2Oasi Research Institute-IRCCS, Via Conte Ruggero 73, 94018 Troina, Italy; 3Institute for Biomedical Research and Innovation (IRIB), National Research Council of Italy (CNR), 98100 Messina, Italy; 4Division of Child Neurology and Psychiatry, Department of the Adult and Developmental Age Human Pathology, University of Messina, 98122 Messina, Italy; 5Department of Clinical and Experimental Medicine, University of Messina, 98125 Messina, Italy; 6Division of Child Neurology and Psychiatry, Federico II University Hospital Naples, 80138 Naples, Italy; 7Department of Psychology, Educational Science and Human Movement, University of Palermo, 90144 Palermo, Italy; 8Child Psychiatry, Neurology Unit, Department of Health Sciences, University of Catanzaro, 88100 Catanzaro, Italy

**Keywords:** autism spectrum disorder, early intervention, early start denver model, early intensive behavioral intervention, treatment outcome

## Abstract

Autism spectrum disorder is a neurodevelopmental disorder with a rising prevalence disorder. This high-cost/high-burden condition needs evidence-based behavioral treatments that are able to reduce the impact of symptoms on children’s functioning. This retrospective chart review study compared the impact of different types of early interventions on toddlers diagnosed with an autism spectrum disorder developmental profile. Analyses were conducted on 90 subjects (mean = 27.76 months, range 18–44 months; M:F = 4.29:1), of which 36 children underwent the usual treatment, 13 children underwent an intervention based on early intensive behavioral intervention (EIBI) and 41 children received the Early Start Denver Model, for one year, with the same weekly frequency of about 6 h a week. A significant decrease in the severity of autism symptoms was observed for all children when looking at the Ados-2 severity score (average difference = 3.05, SD = 0.71, *p* = < 0.001) and the Ados-2 social subscale (average difference = 2.87, SD = 0.59, *p* < 0.001). Otherwise, for most of the Griffiths subscales, we found a significant improvement only for those children who underwent the Early Start Denver Model intervention (General Quotient average difference = 14.47, SD = 3.22, corrected *p* < 0.001). Analyzing the influence of age on the investigated scores, we found a significant association with the Eye–hand Coordination Quotient (*p* = 0.003), Performance Quotient (*p* = 0.042) and General Quotient (*p* = 0.006). In all these domains, a mild negative correlation with age was observed, as measured by the Pearson’s correlation coefficient (r = −0.32, *p* = 0.002; r = −0.21, *p* = 0.044; r = −0.25, *p* = 0.019, respectively), suggesting less severe developmental skills at the start of treatment for older children. Our results are consistent with the literature that underlines the importance of early intervention, since prompt diagnosis can reduce the severity of autism symptoms; nevertheless, in toddlers, our study demonstrated that an intervention model based on naturalistic developmental behavioral principles such as the Early Start Denver Model is more effective on children’s developmental profile. Further studies are required to assess the extent of effectiveness of different early intervention models in community settings.

## 1. Introduction

Autism spectrum disorder (ASD) is a neurodevelopmental condition considered to be one of the major causes of disability in children under 5 years of age [1]. The number of children diagnosed with ASD has steadily increased over the past two decades, with an estimated prevalence of 1 out of 54 individuals in the USA [2] and 1 out of 77 in Italy [3]. The clinical presentation of ASD is frequently associated with intellectual disability and other neurodevelopmental disorders, such as attention deficit hyperactivity disorders and specific motor and language disorders [4]; prognosis has important implications for families, and it is considered a serious public health concern, including the economic aspects [5,6]. Contrary to the past, children with ASD are now being diagnosed as early as the age of 2 years [7,8] and identified as at-risk for ASD between 12 and 24 months of age [9], but early diagnosis is not always followed by an appropriate, specialized intervention. Many behavioral early interventions have been described as effective on ASD core symptoms, such as the autism severity [10,11,12,13]. However, a significant percentage of children continue to show delays in one or more neurodevelopmental areas [14]. Furthermore, controlled trials mainly come from highly specialized centers, involving high levels of staff training and supervision and high intensity services. Currently, current priorities in ASD research include efficacy and effectiveness studies on community setting intervention [15]. Over the last few years, a limited number of studies have shown evidence-based results of early intervention models within community settings [15,16,17]. Despite efforts, documenting implementation strategies utilized in community settings remains a challenge and, in many geographic areas, children with ASD receive non-specialized treatments. For instance, in Europe there is a considerable variability in types and intensity of interventions, and in some countries more than 20% of children receive no intervention at all; public treatment centers are rarely specialized in autism, and the most frequently reported treatments are speech and language therapy (64%) and behavioral, developmental and relationship-based treatments (55%) [18,19]. Therefore, public early intervention in ASD differs with regard to the underlying developmental theory, treatment strategies, intervention targets, involvement of parents and intervention intensity, and thus it is heterogeneous and poorly evaluated [20,21]. In particular, in our study, treatment as usual included autism psychoeducation and ASD impairment core areas-focused intervention.

In the present study, we report the results of a systematic retrospective chart review involving toddlers who received a first diagnosis of either ASD or severe risk for ASD (<30 months) in our clinical units and following the diagnosis started an early intervention with different types of treatment, available from local child neuropsychiatric services in their living area for a time window of one year. 

We took in account three early intervention treatments delivered by community providers, which include: the Early Start Denver Model (ESDM), early intensive behavioral intervention (EIBI) and treatment usually provided in Italy (treatment as usual, TAU). ESDM is a naturalistic developmental behavioral intervention (NDBI) that integrates relationship-based pivot response treatment (PRT; [22,23]) with applied behavior analysis (ABA; [24]). This early intervention, addressed for children with ASD aged 12 to 48 months, is focused on early interaction and social motivation in the context of active experimental learning. Additionally, the frame of this activity is the joint activity routines, designed to mimic the social exchange that takes place in the early relationship between children and caregivers [25]. EIBI is a highly structured teaching approach that is rooted in the principles of ABA [26,27]. It is designed for ASD children younger than 5 years of age. The theoretical foundation of EIBI includes a specific teaching procedure referred to as discrete trial training, initially delivered in a one-on-one highly structured setting. Typically, EIBI procedures follow a specific sequence of tasks, defined by a treatment manual [26,28], and instruction is transferred to both school and home settings to promote generalization and maintenance. Although both early interventions are rooted in ABA by their developers, the ESDM model differs from EIBI in the use of developmental constructs, characterized by social aspects such as positive affect, sensitivity, responsivity and a joint activity routines format [29], despite ABA practice’s increasingly naturalistic approaches [30]. Finally, treatment as usual is a non-specialized treatment provided by the Italian public health system, typically composed of speech and neuropsychomotor therapy. These are characterized by the use of a 1:1 adult-to-child ratio, the non-observance by therapists of a structured program and child-neuropsychiatrist-defined and reviewed intervention objectives and strategies.

To our knowledge, this is the first study comparing the impact of three types of autism early interventions (as usual, Early Start Denver Model, ESDM, or early intensive behavioral intervention, EIBI) administered in community contexts on developmental profiles. Although there are several studies showing that ESDM and EIBI are effective treatments for young children with ASD, they were conducted comparing each target intervention to a no-treatment or treatment-as-usual control condition. 

There are also comprehensive systematic reviews and meta-analysis studies reporting the efficacy of different nonpharmacological early interventions for ASD, but, unlike our study, the large samples considered had a rather broad age target, and they did not specifically analyze the first years of development [31].

This study aimed to verify the efficacy of evidence-based interventions in community settings to ensure that the models developed in university settings are feasible and sustainable in services not supported by research funds. This systematic retrospective chart review aims to provide evidence of the impact on developmental profile of three different types of early intervention programs (treatment as usual, Early Start Denver Model or early intensive behavioral intervention) delivered in community contexts, without rigorous research control. Effectiveness was measured by comparing the impact of intervention on developmental profile and autism core symptoms of toddlers diagnosed as ASD or severe risk for ASD.

## 2. Materials and Methods

We performed a retrospective chart review of all cases aged between 18 and 44 months, consecutively referred to the Child and Adolescent Psychiatry Unit of the “G. Martino” University Hospital in Messina (Italy) between 1 January 2014 and 1 December 2018. 

We selected charts of all children who received a first diagnosis of either ASD or severe risk for ASD at age <30 months. Diagnoses were made by experienced child and adolescent psychiatrists, according to the Diagnostic and Statistical Manual of Mental Disorders 5th ed. [32] based on direct observation, parental reports and using standard valid and reliable assessment tools. This standard panel of psychodiagnostics tests included a measure of autism symptoms (Autism Diagnostic Observation Schedule, Second Edition—ADOS-2, Italian version) [33,34] administered by experienced clinicians trained for research reliability. Furthermore, a measure of the children’s level of development (Griffiths Mental Developmental Scales—Extended Revised; GMDS-ER) [35] was administered by senior psychologists experienced in clinical evaluations. 

We took into account the following inclusion criteria: (i) none of the children had received any kind of intervention prior to diagnosis; (ii) all participants started an early intervention based on one of the following treatments: treatment as usual (TAU), Early Start Denver Model (ESDM) or early intensive behavioral intervention (EIBI) for one year, with the same weekly intensity (mean 6 ± 1 h per week); (iii) a follow-up visit after one year of treatment completed by a standard panel of psychodiagnostics tests. All subjects who presented other significant medical conditions (e.g., epilepsy, significant hearing and visual sensory deficits, traumatic brain injury or other significant genetic disorders) or who simultaneously underwent pharmacological treatment were excluded. 

Baseline and endpoint measures of developmental trajectory and ASD symptoms severity during the follow-up were as follows: (i) the GMDS-ER is a standardized developmental test used in numerous studies conducted by researchers on the Italian population. The GMD-ER evaluates different areas of functioning in young children, comprising a score in five subscales: Locomotor (assessing gross motor skills including the ability to balance, coordinate and control movements), Personal Social (measuring the developing abilities that contribute to independence and social development), Hearing and Language (assessing receptive and expressive languages), Eye and Hand Coordination (focuses on fine motor skills, manual dexterity and visual monitoring skills) and Performance (assessing the child’s visuospatial skills, speed of working and precision). The subscale quotients are calculated using the developmental age corresponding to each subscale divided by the actual chronological age and multiplied by 100. The mean of the General Quotient (GQ) and each of the six subscale quotients is 100 points (SD = 15 points). The GQ raw score is the sum of the subscales raw scores. A GQ or a subscale quotient <70 points (>2SD below the mean) is considered indicative of developmental delay. (ii) The ADOS-2—a semi-structured observation tool measuring symptoms of autism in social communication, play and repetitive behaviors—provides an empirically derived algorithm that differentiates children with ASDs from those with other delays or typical development. Furthermore, it separately measures two main aspects: social affect (SA) and restricted, repetitive behaviors (RRBs). The ADOS-2 consists of different modules with an activity program designed for children with different levels of language development. The ADOS calibrated severity score was calculated by the assessment, and the choice of the appropriate ADOS module was based on each child’s language level [36,37]. 

The data retrieved from chart review included child’s sex, age, DSM-5 diagnosis, main symptoms, kind and frequency of treatment performed—as recorded by the clinician following each patient—and pre-post intervention psychological assessments. Written informed consent for the use for scientific and publication purposes of data derived from clinical children work-up was collected from both parents and legal guardians of all patients. 

### Statistical Analyses

To understand how the different intervention protocols influenced toddlers’ development, GMDS-ER subscales were considered as primary outcome measures. The secondary outcome measure was the ADOS-2 severity score. These two psychodiagnostics tests, sampled at the onset of the treatments (T0) and after 1 year (T1), were analyzed.

Due to the nature of the study, a repeated measure approach was adopted: to account for potential violations of standard repeated measure ANOVA, a Pillai multivariate test was used instead. In the analysis, treatment (TAU, ESDM or EIBI) was included as between-subjects factor, timepoint (T0 vs. T1) factor was included as within-subjects factor and age was included as covariate. In case of significant main factors or interactions, Bonferroni correction has been adopted for post hoc comparisons; such a conservative choice was made to show results as robustly as possible. Overall significance threshold was set to 0.05.

## 3. Results

A total of 90 subjects satisfied the study selection criteria. The demographic and clinical characteristics of the sample are summarized in Table 1. The age range was 18–44 months (mean = 27.76 months, SD = 5.69 months). Among them, 73 males and 17 females were recruited, reproducing a sex ratio of about M:F = 4.29:1. This sex ratio is in line with an excess of males being affected by ASD. Based on clinical charts reviews, 36 patients were treated with TAU, 41 ESDM and 13 EIBI. No significant differences were observed for the General Quotient ADOS severity score at T0. 

### 3.1. Developmental Domains

After one year of treatment, for most of the Griffiths subscales, we found a significant increase at T1 only for toddlers who underwent ESDM, whereas no significant change was found for neither TAU nor EIBI ones (Figure 1). This result was observed in the Locomotor Development subscale (average difference = 22, SD = 6.3, corrected *p* = 0.008), Personal Social Development (average difference = 24.37, SD = 5.27, corrected *p* < 0.001), Hearing and Speech (average difference = 30.80, SD = 5.15, corrected *p* < 0.001), Hand and Eye Coordination (average difference = 16.59, SD = 5.3, corrected *p* = 0.024) and GQ (average difference = 14.47, SD = 3.22, corrected *p* < 0.001) (Table 2).

When analyzing the influence of age on the investigated scores, we found a significant association with Hand and Eye Coordination (*p* = 0.003), Performance Test (*p* = 0.042) and GQ (*p* = 0.006). On all those occasions, a mild negative correlation was observed with age as measured via Pearson’s correlation coefficient (r = −0.32, *p* = 0.002; r = −0.21, *p* = 0.044; r = −0.25, *p* = 0.019, respectively), meaning that lower scores were observed at T0 for older toddlers. For detailed statistical results, please refer to Table 3.

### 3.2. Autism Severity

One year after treatment, all three different groups significantly improved in terms of autism severity core symptoms. In particular, a significant decrease at T1 was observed, on average, for all treatment groups when looking at the ADOS severity score (average difference = 3.05, SD = 0.71, *p* ≤ 0.001) and AS subscale (average difference = 2.87, SD = 0.59, *p* < 0.001).

Conversely, the repetitive behavior scores did not change over time in either group. No other significant differences were observed regarding autism severity.

## 4. Discussion

The purpose of this study was to evaluate the impact of different kinds of early interventions on toddlers who received an early diagnosis of ASD in terms of severity of autistic symptoms and developmental profiles. To our knowledge, this is the first study comparing the impact of three types of autism early interventions (as usual, ESDM and EIBI), administered in community contexts, on developmental profiles. Although there is no unanimous international consensus on when to perform the first neuropsychiatric evaluation in the case of suspected ASD, a widely accepted period is within 18 months [38].

Therefore, it is extremely important to clearly define the most appropriate early interventions for toddlers with ASD. In our study, one year after treatment, all the three different early interventions (as usual, ESDM and EIBI) similarly significantly improved the severity of autism core symptoms. This seemingly surprising result is compatible with other similar evidence in the literature showing how individual therapy offered by local services, and including sessions of individual psychomotricity and/or speech therapy and/or psychoeducative therapy, together with school-supported inclusion, could determine improvements in the severity of autism core symptoms, cognitive and linguistic development, adaptive behaviors and comorbid psychopathology [39].

The personalization of therapeutic interventions, parental involvement with the recognition of the child’s cues and starting therapy early and at a sufficient intensity that is of an appropriate duration are likely the reasons for the success of this approach. The aforementioned improvements in ASD core symptoms were not matched with similar improvements in cognitive profiles. A significant increase in four out of the five Griffiths subscales (Locomotor Development, Personal Social Development, Hearing and Speech and Hand and Eye Coordination) was observed only for toddlers who underwent ESDM. This observation confirms the results showed by previous scientific literature underlying significant improvements in IQ, adaptive behavior and diagnostic status in children who received ESDM treatment early compared with children who received community interventions [12].

This data suggests that ESDM treatment is able to support the global development of children with ASD even when it is practiced outside a research context and therefore without the certainty of a standardized intervention and a supervised healthcare team.

The EIBI group does not highlight significant improvements in developmental profiles despite the use of applied behavior analysis principles. A recent meta-analysis by Reichow et al. considered five studies with a total of 219 children: 116 children in EIBI groups and 103 children in TAU groups, who provided weak evidence that EIBI improves adaptive behavior, IQ, expressive and receptive language, everyday communication skills, everyday social competences, daily living skills and problem behavior as well as the autism symptom severity [40].

Therefore, our results are in contrast with the evidence in the literature, and this incongruence is probably due to the low number of cases in the EIBI group (13), representing a limitation of the study to point out. 

However, the present study has some limitations to report. We acknowledge that the three groups we investigated had different numbers of participants, and the unequal sample size may have affected statistical power [41]. The consequent power loss may have potentially masked effects of interests, which we were instead able to find, thus proving the strength of our results. To furthermore limit potential related issues, we adopted a Bonferroni correction for post hoc analysis, which is recognized as the most conservative approach. For the above reasons, we are confident about the consistency of our findings. 

Another weakness of our study is the closeness of the two compared approaches, ESDM and EIBI, and the lack of precise descriptions of TAU across the various fidelity monitoring of TAU intervention.

However, TAU was representative of the treatment usually provided in Italy, consisting of psychomotor and language skills education. The several major limitations of our study are in its experimental design. In fact, this was a retrospective study based on medical charts, which have a lower intrinsic level of evidence compared with prospective studies [42]. Nonetheless, in our opinion, this type of research design is a valuable tool with which to evaluate the impact on the developmental profile of early treatment available in community settings to children with ASD attending services not supported by research funds. Despite these limitations, the study provides useful information for the management of toddlers and children with ASD.

## 5. Conclusions

The outcomes of this study, which involved a significant decrease in the severity of autistic symptoms for all patients early treated with treatment as usual or the Early Start Denver Model or early intensive behavioral intervention, consistently overlap with other studies on intensive early interventions in children with ASD. 

The results of this study confirm once again the importance of an early diagnosis and underline how timely and intensive interventions are more critical in determining improvements in the outcome in people with ASD rather than the type of intervention used.

Our retrospective study provided encouraging results, supporting the efficacy of early intervention, since prompt diagnosis could reduce the severity of autism symptoms; nevertheless, in toddlers, it seems that an intervention based on Early Start Denver Model principles, without rigorous research control, may be more effective for improving children’s developmental profiles. Randomized controlled trials will be necessary to confirm these observations, verifying both efficacy and treatment response of early treatment available in community settings, to better assess the need for implementation and standardization of the offer in the public health system.

## Figures and Tables

**Figure 1 jcm-11-05424-f001:**
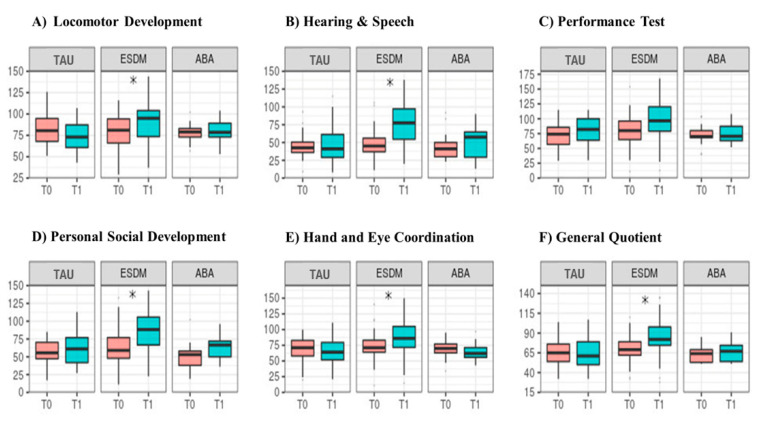
Developmental profile after one year of treatment. Scores in the (**A**) Locomotor Development subscale, (**B**) Hearing and Speech s., (**C**) Performance Test, (**D**) Personal Social Development s., (**E**) Hand and Eye Coordination s. and (**F**) General Quotient scale of the ESDM group, which increased after treatment in comparison to the TAU and EIBI group. * indicate significant differences between groups.

**Table 1 jcm-11-05424-t001:** Baseline demographic and clinical characteristics of the sample (N = 90).

Demographic and Clinical Characteristics	
Age in months (mean ± s.e.m. and range)	27.76 ± 5.69 (18–44)
Gender (Male/Female)	73/17
M:F ratio	4.29:1
General Quotient (mean ± s.e.m.)	
TAU group	65.34 ± 16.57
ESDM group	69.90 ± 17.08
EIBI group	63.46 ± 10.24

TAU: Treatment As Usual; ESDM: Early Start Denver Model; EIBI: Early Intensive Behavioral Intervention.

**Table 2 jcm-11-05424-t002:** Significant differences in developmental profile after one year of treatment.

Effects	Avg. Diff.	TAU	Corrected *p*
Personal Social Development			
TAU T0 vs. ESDM T1yr	−29.37	6.00	<0.001
ESDM T0 vs. ESDM T1yr	−24.37	5.27	<0.001
EIBI T0 vs. ESDM T1yr	−33.48	8.35	0.0013
TAU T1yr vs. ESDM T1yr	−25.18	6.01	<0.001
Hearing and Speech			
TAU T0 vs. ESDM T0	−33.15	6.17	<0.001
ESDM T0 vs. ESDM T1yr	−30.80	5.15	<0.001
EIBI T0 vs. ESDM T1yr	−32.28	8.58	0.0031
TAU T0 vs. ESDM T1yr	−29.46	6.18	<0.001
Hand and Eye Coordination			
TAU T0 vs. ESDM T1yr	−19.76	6.02	0.015
ESDM T0 vs. ESDM T1yr	−16.59	5.3	0.024
TAU T1yr vs. ESDM T1yr	−23.36	6.03	0.002
TAU T0 vs. ESDM T1yr	−19.76	6.02	0.015
General Quotient (GQ)			
TAU T0 vs. ESDM T1yr	−17.77	4.40	0.001
ESDM T0 vs. ESDM T1yr	−14.47	3.22	<0.001
EIBI T0 vs. ESDM T1yr	−19.50	5.98	0.016
TAU T1yr vs. ESDM T1yr	−22.67	4.40	<0.001

**Table 3 jcm-11-05424-t003:** Statistical results of retrospective analyses of ADOS and Griffiths scores and subscales.

Dependent Variable	Treatment (TAU, ESDM, EIBI)	Timepoint (T0 vs. T1)	Treatment * Timepoint Interaction	Age and Correlation Analysis
ADOS-2 Severity Score	*p* > 0.05	*p* = 5.02 × 10^−^^5^ Post hoc: decrease at T1: (3.05, 0.71, *p* < 0.001) *	*p* > 0.05	*p* > 0.05
ADOS-2 Social Affect	*p* > 0.05	*p* = 5.31 × 10^−6^ Post hoc: decrease at T1: (2.87, 0.59, *p* < 0.001)	*p* > 0.05	*p* > 0.05
ADOS-2 Restricted/repetitive Behaviors	*p* > 0.05	*p* > 0.05	*p* = 0.04 Post hoc: any significance after Bonferroni correction	*p* > 0.05
GMDS-ER Locomotor Development	*p* > 0.05	*p* > 0.05	*p* = 0.014 Post hoc: ESDM group significant increase at T1 (22.00, 6.3, *p* = 0.008)	*p* > 0.05
GMDS-ER Personal Social Development	*p* = 0.006 Post hoc: ESDM increased score over TAU (15.09, 4.61, *p* = 0.004)	*p* = 4.34 × 10^−5^ Post hoc: increase at T1 (17.35, 4.03, *p* < 0.001)	*p* = 0.026 Post hoc: ESDM group significant increase at T1 (24.37, 5.27, *p* < 0.001)	*p* > 0.05
GMDS-ER Hearing and Speech	*p* = 0.006 Post hoc: ESDM increased score over TAU (15.90, 4.90, *p* = 0.004)	*p* = 0.0001 Post hoc: increase at T1 (15.86, 3.93, *p* = 0.0001)	*p*= 0.002 Post hoc: ESDM group significant increase at T1 (30.80, 5.15, *p* < 0.001)	*p* > 0.05
GMDS-ER Hand and Eye Coordination	*p* = 0.02 Post hoc: ESDM increased score over TAU (13.26, 4.62, *p*= 0.013)	*p* > 0.05	*p* = 0.037 Post hoc: ESDM group significant increase at T1 (16.59, 5.3, *p* = 0.024)	*p* = 0.003 Pearson’s correlation analysis (r = −0.32, t = −3.17, df = 88, *p* = 0.002);
GMDS-ER Performance Test	*p* = 0.040 Post hoc: ESDM increased score over TAU (13.45, 5.24, *p*= 0.031)	*p* = 0.002 Post hoc: increase at T1(13.98, 4.41, *p* = 0.0021)	*p* > 0.05	*p* = 0.042 Pearson’s correlation analysis (r = −0.21, t = −2.04, df = 88, *p* = 0.044)
GMDS-ER General Quotient	*p* = 0.002 Post hoc: ESDM increased score over TAU (12.98, 3.69, *p*= 0.002)	*p* > 0.05	*p* = 0.0005 Post hoc: ESDM group significant increase at T1 (14.47, 3.22, *p* < 0.001)	*p*= 0.006 Pearson’s correlation analysis (r = −0.25, t = −2.40, df = 88, *p* = 0.019)

* In case of significant results, post hoc *p*-values are intended as Bonferroni-corrected, as stated in the main body of the manuscript. In case of significant results, values between parentheses indicate average difference, standard deviation and corrected *p* as estimated by the model.

## Data Availability

The data that support the findings of this study are available from the corresponding author upon reasonable request.

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
