# Peer review of "Impact of Three Kinds of Early Interventions on Developmental Profile in Toddlers with Autism Spectrum Disorder"

_jcm, 2022, doi:10.3390/jcm11185424_

Round 1
Reviewer 1 Report
This is an interesting retrospective study showing that the behavioral ASD symptoms, if diagnosed early and treated, can be improved by treatment. The authors also show that Early Start Denver Model of ealy intervention seemed to be superior over the method using the principles of Applied Behavior Analysis (Early Intensive Behavioral Intervemtion -EIBI) or regular neurodevelopmental intervention (TAU). The data are interesting although the number of children who had EIBI is small, and the conclusions based on this number are somewhat vague. I think, however, that the authors need to better define the group of children with ASD. What were the criteria for the determination of children with "severe risk for ASD"? What was the minimal ADOS score to define these children or the children with ASD? How many children were with "severe risk for ASD" and how many with ASD in each group? For proper assessment of the results all these details are improtant.
Author Response
Dear Reviewer,
I would like to thank you for your valued comments and suggestions to the article. As you requested, we made all the necessary changes in our manuscript to address your concerns and we detailed below how the points raised have been accommodated. From the changes made in the revised manuscript and responses provided below, I hope you are convinced that we have adequately addressed the reviewer’s concerns and made the paper better. If there are any further questions, please feel free to let me know.
REVIEWER 1
This is an interesting retrospective study showing that the behavioral ASD symptoms, if diagnosed early and treated, can be improved by treatment. The authors also show that Early Start Denver Model of early intervention seemed to be superior over the method using the principles of Applied Behavior Analysis (Early Intensive Behavioral Intervemtion -EIBI) or regular neurodevelopmental intervention (TAU). The data are interesting although the number of children who had EIBI is small, and the conclusions based on this number are somewhat vague. I think, however, that the authors need to better define the group of children with ASD. What were the criteria for the determination of children with "severe risk for ASD"? What was the minimal ADOS score to define these children or the children with ASD? How many children were with "severe risk for ASD" and how many with ASD in each group? For proper assessment of the results all these details are important.
We wish to thank you for expressing interest in our results and for helping us to improve our manuscript. We have modified Materials and Methods paragraph providing more details, and we added new citations to better clarify the decision-making process inherent in ADOS2 modules (line 154). We verify the absence of significant differences for ADOS severity score and include this information in Result paragraph.
Reviewer 2 Report
This is an exciting topic and deserves more attention from the clinical perspective. Hence, the manuscript authors need to consider the following points:
1. In the study title, mentioning or naming the evaluated approaches is essential. Therefore, the present title needs to be amended to cover that three or two approaches are investigated, not the whole approaches available for early intervention.
2. It is also necessary to introduce the adopted approaches from a clinical perspective and present the information regarding each approach's similarities and differences. Presently only Denver and Behavioural models are presented.
3. On lines 93 to 95, when it is mentioned the aim and objectives of the study it needs to be clear and specific and name the approaches that have been considered and investigated in the study.
4. No conceptual models were adopted for the study. What was the perspective of the researcher? What was necessary for performing such an analysis? What type of gap was noticed in the available literature, and answering all these questions reveals the conceptual and theoretical perspective of the present study that has been missed at the moment.
5. To be able to perform the ADOS 2 special certificate is needed, make it clear if the professionals who used the scale were certified or not, and also it will be conducive to indicate that the recruited children had already received the diagnosis by means of ADOS2 or it was done because of the present study.? Was an Italian version of the scale used or the original English scale.
6. Make it clear which modules of ADOS2 were used and what was the recruited sample's level of ASD severity based on the ADOS2 scale
7. Is it true that only one experienced Child & Adolescent Psychiatrist completed all the diagnoses? What was the reason for that when the sample had already received a diagnosis using ADOS 2?
8. It is very helpful to mention the inclusion criteria. Still, I am sure that the approach mentioned as the treatment as usual TAU needs to be introduced and presented and defined in the introduction part. I am very puzzled that TAU as an independent approach was mentioned without providing a clear definition that separates it from the two other approaches.
9. You should need to explain what you mean by receiving no other types of early intervention services as an inclusion criterion ( on lines 111 to 112: none of the children had received any kind of intervention prior to diagnosis). How did you control that? Was it ethical? Why did you not adopt an RTC in which you could match children who received different early intervention services?
10.On line 116, it is said that the recruited sample was re-evaluated using a standard panel of psychodiagnostics tests. This panel needs to be explained in more detail here.
11.Make it clear how many children were approached and how many were excluded because of the inclusion criteria.
12.Regarding the baseline and final evaluation scales GMDS-ER and ADOS2, more information is needed (especially the GMDS-ER, the used version, and its applicability ( validity and reliability ) with the Italian population).
13.I also suggest using general terms such as client, children with ASD, or similar terms instead of the "patient" as used in line 142.
14.Why have different terms been used to indicate the used approaches? In the text, TAU, ESDM, and EIBI are used, while STD, ESDM, and ABA are used in the figure. I suggest using similar terms in the entire text.
15.I think the limitation of the present study goes far beyond what has presently provided. To me, the most important shortcoming of the study is its methodological approach; instead of an RCT (Randomized controlled trials, a pre and post-approach is considered. RCTs are prospective studies that measure the effectiveness of an intervention or treatment. Although no study is likely on its own to prove causality, randomization reduces bias and provides a rigorous tool to examine cause-effect relationships between an intervention and outcome. I suggest reading a citing the following very important review study:
Yang, Y. H. (2019). Review of early intervention for children with autism spectrum disorder: focused on randomized controlled trials. Journal of the Korean Academy of Child and Adolescent Psychiatry, 30(4), 136.
16.I also think another scale needed to be administered instead of diagnostic scales such as ADOS2 to determine the impacts of the interventions. Some scales such as ATEC ( Autism Treatment Evaluation Checklist). A major obstacle in autism research has been the lack of a valid means of measuring the effectiveness of various treatments. And the application of the diagnostic scales is not recommended for the validity of the intervention approaches. Over the years, researchers have published hundreds of studies attempting to evaluate different biomedical and psycho-educational interventions intended to benefit children with autism. Much of this research produced inconclusive or, worse, misleading results because there are no useful tests or scales designed to measure treatment effectiveness. Lacking such a scale, researchers resorted to using scales such as ADOS ( as it was used in this study) or other scales such as the Childhood Autism Rating Scale (CARS), the Gilliam Autism Rating Scale (GARS), or the Autism Behavior Checklist (ABC), all of which were designed to diagnose autism- to tell whether or not a child has ASD--and not to measure treatment effectiveness.
I agree that this is very important to understand the impacts of different intervention approaches, but I also think such studies are very difficult to administer.
I also think that considering the Denver model as a different approach from the Behavioral model is not a very strong premise. The Early Start Denver Model (ESDM) is a behavioral therapy for children with autism between the ages of 12-48 months. This approach is based on the methods of applied behavior analysis (ABA). I also think that the other unclear approach in your study, AUT, might have the ABA components in its intervention. This is a very strong justification for having no statistically significant difference between the three groups in your study. I also think that comparing ESDM with the other nonbehavioral models, such as the Hanen approach, might yield different results. Hanen programs reflect a family-centered model of intervention based on a social-interactionist perspective of language development. The Hanen approach is based on the principles that parents should be involved in their child's intervention process and that communication should be facilitated in naturalistic contexts.
Parents and therapists use play to build positive and fun relationships. Through play and joint activities, the child is encouraged to boost language, social and cognitive skills.
You could also compare clinical-based and home-based approaches to early intervention to understand the possible differences between different methods with different conceptual bases.
In sum, I think that your study used diverse terminology (ESDM, TAU, and EIBI); in reality, they were possible all the same, and the results support this justification. Using different words to express similar meanings can lead to difficulties understanding and implementing the interventions in detail. Particularly no data has been presented at all regarding TAU, and children who received this approach also had no statistically significant difference from the other two groups. This is very crucial to know what was the contents of this approach that was this effective. In particular, you were not indicated which approaches were implemented directly or indirectly by parents or caregivers, who are non-experts, and which approach was pure clinical and presented by the clinicians. This could be an even more significant obstacle to generalizing the present findings. As the Early intervention services provider or researcher, you need to classify and define interventions using standard, clearly, widely understood terms (Behavioral, cognitive, communication, or parental or clinicians based). In the case of interventions that do not belong to an existing classification (in your case, TAU), it will be necessary to introduce the principles and specific implementation methods in more detail.
Author Response
Dear Reviewer,
I would like to thank you for your valued comments and suggestions to the article. As you requested, we made all the necessary changes in our manuscript to address your concerns and we detailed below how the points raised have been accommodated. From the changes made in the revised manuscript and responses provided below, I hope you are convinced that we have adequately addressed the reviewer’s concerns and made the paper better. If there are any further questions, please feel free to let me know.
Sincerely,
Luigi Vetri,
REVIEWER 2
- In the study title, mentioning or naming the evaluated approaches is essential. Therefore, the present title needs to be amended to cover that three or two approaches are investigated, not the whole approaches available for early intervention.
We provide to modify the study title, indicating the number of different approaches evaluated, as follow:
“Impact of three kinds of early interventions on develop-mental profile in toddlers with autism spectrum disorder”
- It is also necessary to introduce the adopted approaches from a clinical perspective and present the information regarding each approach's similarities and differences. Presently only Denver and Behavioural models are presented.
In lines 72-78 we provide a literature description about public treatment, and below we have added more details about treatment-as-usual group in our study.
- On lines 93 to 95, when it is mentioned the aim and objectives of the study it needs to be clear and specific and name the approaches that have been considered and investigated in the study.
Done, we provide to specify the name of approaches take in account in this study.
- No conceptual models were adopted for the study. What was the perspective of the researcher? What was necessary for performing such an analysis? What type of gap was noticed in the available literature, and answering all these questions reveals the conceptual and theoretical perspective of the present study that has been missed at the moment.
Early diagnosis and screening to ensure early intervention is a recent goal to promote positive outcomes for children and families. There is a growing body of evidence from randomized trials reporting promising results on the efficacy of early and intensive behavioral intervention (Narzisi et al., 2013; Yang et al., 2019); also, there are many national guidelines for autism that recommend early specialist interventions adopted by several countries. Quite the opposite, little is known about the effectiveness on developmental profile of interventions delivered by public assistance centers, which are rarely specialized for ASD. Evaluating evidence-based interventions in community settings is currently a step needed in the intervention research realm to ensure that the models developed in university settings are feasible and sustainable in services not supported by research funds. We agree that this study is not designed to provide a broad and definitive answer to those questions. However, this systematic retrospective chart review (line 81) would verify the impact on developmental profile of three different kind of early treatment delivered from local child neuropsychiatric services in their living area.
To explain better this experimental question, the final part of introduction were rewritten providing more details, as follows:
“The aim of our study would verify the efficacy of evidence-based interventions in community settings to ensure that the models developed in university settings are feasible and sustainable in services not supported by research funds. This systematic retrospective chart review is to provide evidence of the impact on developmental profile of three different types of early intervention programs (treatment as usual, Early Start Denver model or early intensive behavioral intervention) delivered in community contexts, without rigorous research control. Effectiveness was measured by comparing the impact of intervention on developmental profile and autism core symptoms of toddlers diagnosed as ASD or severe risk for ASD.”
- To be able to perform the ADOS 2 special certificate is needed, make it clear if the professionals who used the scale were certified or not, and also it will be conducive to indicate that the recruited children had already received the diagnosis by means of ADOS2 or it was done because of the present study.? Was an Italian version of the scale used or the original English scale.
- ADOS 2 special certificate indicated in line 120. We added the reference to Italian version [25]
- We agree with the view expressed by the Reviewer. Initial part of Materials and Methods has been rewritten providing more details, as follows:
“We performed a retrospective chart review of all cases aged between 18-44 months, consecutively referred to the Child and Adolescent Psychiatry Unit of the “G. Martino” University Hospital in Messina (Italy) between January 1, 2014 and December 1, 2018.
We selected charts of all children who received a first diagnosis of either ASD or severe risk for ASD at age<30 months. Diagnosis were made by experienced Child & Adolescent Psychiatrist, according to the Diagnostic and Statistical Manual of Mental Disorders 5th ed. [23] based on direct observation, parental report and using standard valid and reliable assessment tools. This standard panel of psychodiagnostic tests included a measure of autism symptoms (Autism Diagnostic Observation Schedule, Second Edition - ADOS-2, Italian version) [24] administered by experienced clinicians trained for research reliability, and a measure of the children’s level of development (Griffiths Mental Developmental Scales-Extended Revised; GMDS-ER) [25], administered by senior psychologists experienced in clinical evaluations.“
- Make it clear which modules of ADOS2 were used and what was the recruited sample's level of ASD severity based on the ADOS2 scale
We have modified Materials and Methods paragraph providing more details, and we added new citations to better clarify the decision-making process inherent in ADOS2 modules (line 154). We verify the absence of significant differences for ADOS severity score and include this information in Result paragraph.
- Is it true that only one experienced Child & Adolescent Psychiatrist completed all the diagnoses? What was the reason for that when the sample had already received a diagnosis using ADOS 2?
We provide to improve initial part of Materials and Methods, that has been rewritten (“Diagnosis were made by experienced Child & Adolescent Psychiatrists”). In Italian public system, diagnosis were made by Child & Adolescent psychiatrist, according to Nice guideline (Autism spectrum disorder in adults: diagnosis and management, National Institute for Health and Care Excellence, 2021) and Italian Society of Childhood and Adolescence Neuropsychiatry guidelinee (Linee guida per l’autismo, raccomandazioni tecniche-operative per i servizi di neuropsichiatria dell’eta’ evolutiva, SINPIA, 2005). This processes, team based, involve a family member, carer or other informant or use documentary evidence (such as school reports) of current and past behaviour and early development. To better complete the assessment process, our neuropsychiatric unit use formal assessment tools, such as the ADOS2 and the Griffiths Mental Development scales – extended revised.
- It is very helpful to mention the inclusion criteria. Still, I am sure that the approach mentioned as the treatment as usual TAU needs to be introduced and presented and defined in the introduction part. I am very puzzled that TAU as an independent approach was mentioned without providing a clear definition that separates it from the two other approaches.
Taking into account these suggestions, in lines 72-78 we provide a literature description about public treatment, and below we have added more details about treatment-as-usual group in our study.
- You should need to explain what you mean by receiving no other types of early intervention services as an inclusion criterion ( on lines 111 to 112: none of the children had received any kind of intervention prior to diagnosis). How did you control that? Was it ethical? Why did you not adopt an RTC in which you could match children who received different early intervention services?
This is a retrospective chart review (RCR), a type of research design in which pre-recorded, patient-centered data are used to answer one or more research questions (Vassar, M., & Holzmann, M., 2013). Our systematic retrospective chart review would verify the efficacy of three different kind of early treatment delivered from local child neuropsychiatric services in their living area. We reviewed every medical chart record of our medical unit choosing to consider for this study only patients who receiving no other types of early intervention services as an inclusion criterion. Thus, no discrimination was made between patients in course of treatment, but only an a posteriori re-assessment of medical records. Although RCR is a widely applicable research methodology to direct subsequent prospective investigations, therefore we agree with the viewpoint expressed by the Reviewer and we will consider starting an RTC to better answer this experimental question.
In the Conclusions paragraph (page 8), we now state that: “Randomized controlled trials will now be necessary to confirm these observations, verifying both efficacy and treatment response of early treatment available in community settings, to better assess the need for implementation and standardization of the Public Health System offer”
10.On line 116, it is said that the recruited sample was re-evaluated using a standard panel of psychodiagnostics tests. This panel needs to be explained in more detail here.
We now provide to complete the sentence in this way:
“(iii) a follow-up visit after one year of treatment completed by a standard panel of psychodiagnostics tests, consisting of GMDS-ER and Ados 2.”
11.Make it clear how many children were approached and how many were excluded because of the inclusion criteria.
The sampling technique utilized in this study were a systematic sampling. We reviewed every medical chart of our unit from 2014 to 2018, and only considered patients that satisfy inclusion/exclusion criteria. For this reason, it is very difficult to indicate the exact number of charts excluded, because we have examined a large number of medical records.
12.Regarding the baseline and final evaluation scales GMDS-ER and ADOS2, more information is needed (especially the GMDS-ER, the used version, and its applicability ( validity and reliability ) with the Italian population).
We are grateful for helping us to improve our manuscript; we have modified the reference concerning the test (n.25) with a more appropriate one. In this study we used the Griffiths Mental Developmental Scales, version Extended Revised, as indicated in line 109.
About their applicability with the Italian population, beyond the specific validation of the tool, in the literature there are numerous studies conducted by researchers on the Italian population, which confirm the validity and reliability of GMDS-ER for the Italian population, e.g.:
Colombo P, Nobile M, Tesei A, Civati F, Gandossini S, Mani E, Molteni M, Bresolin N, D'Angelo G. Assessing mental health in boys with Duchenne muscular dystrophy: Emotional, behavioural and neurodevelopmental profile in an Italian clinical sample. Eur J Paediatr Neurol. 2017 Jul;21(4):639-647. doi:10.1016/j.ejpn.2017.02.007. Epub 2017 Mar 24. PMID: 28392227.
Zuccarini M, Guarini A, Iverson JM, Benassi E, Savini S, Alessandroni R,Faldella G, Sansavini A. Does early object exploration support gesture and language development in extremely preterm infants and full-term infants? J Commun Disord. 2018 Nov-Dec;76:91-100. doi: 10.1016/j.jcomdis.2018.09.004. Epub 2018 Sep 28. PMID: 30300842.
Lugli L, Bedetti L, Guidotti I, Pugliese M, Picciolini O, Roversi MF, DellaCasa Muttini E, Lucaccioni L, Bertoncelli N, Ancora G, Gargano G, Mosca F, Sandri F, Corvaglia LT, Solinas A, Perrone S, Stella M; Neuroprem Working Group, Iughetti L, Berardi A, Ferrari F. Neuroprem 2: An Italian Study of Neurodevelopmental Outcomes of Very Low Birth Weight Infants. Front Pediatr. 2021 Sep 13;9:697100. doi: 10.3389/fped.2021.697100. PMID: 34589450; PMCID: PMC8474877.
Muglia P, Filosi M, Da Ros L, Kam-Thong T, Nardocci F, Trabetti E, Ratti E, Rizzini P, Zuddas A, Bernardina BD, Domenici E; Italian Autism Network. The Italian autism network (ITAN): a resource for molecular genetics and biomarker investigations. BMC Psychiatry. 2018 Nov 21;18(1):369. doi:10.1186/s12888-018-1937-y. PMID: 30463616; PMCID: PMC6247619.
Andalò B, Rigo F, Rossi G, Majorano M, Lavelli M. Do motor skills impact on language development between 18 and 30 months of age? Infant Behav Dev. 2022 Feb;66:101667. doi: 10.1016/j.infbeh.2021.101667. Epub 2021 Nov 24. PMID:34837789.
and the last one published few time ago, Pino MC, Donne IL, Vagnetti R, Tiberti S, Valenti M, Mazza M. Using the Griffiths Mental Development Scales to Evaluate a Developmental Profile of Children with Autism Spectrum Disorder and Their Symptomatologic Severity [published online ahead of print, 2022 Jun 28]. Child Psychiatry Hum Dev. 2022;10.1007/s10578-022-01390-z. doi:10.1007/s10578-022-01390-z).
According to reviewer request, we provide more information on line 137 as follow:
“(i) the GMDS-ER is a standardized developmental test used in numerous studies conducted by researchers on the Italian population”
13.I also suggest using general terms such as client, children with ASD, or similar terms instead of the "patient" as used in line 142.
Done.
14.Why have different terms been used to indicate the used approaches? In the text, TAU, ESDM, and EIBI are used, while STD, ESDM, and ABA are used in the figure. I suggest using similar terms in the entire text.
Done.
15.I think the limitation of the present study goes far beyond what has presently provided. To me, the most important shortcoming of the study is its methodological approach; instead of an RCT (Randomized controlled trials), a pre and post-approach is considered. RCTs are prospective studies that measure the effectiveness of an intervention or treatment. Although no study is likely on its own to prove causality, randomization reduces bias and provides a rigorous tool to examine cause-effect relationships between an intervention and outcome. I suggest reading a citing the following very important review study:
Yang, Y. H. (2019). Review of early intervention for children with autism spectrum disorder: focused on randomized controlled trials. Journal of the Korean Academy of Child and Adolescent Psychiatry, 30(4), 136.
We agree with the view expressed by the Reviewer, an RCT design is regarded as the 'gold standard' for evaluating the effectiveness of interventions. The methodological limitations of our study are now presented and discussed in much greater detail (line 294), especially its retrospective design, as follows:
“The several major limitation of our study is its experimental design. In fact, this is a retrospective study based on medical charts, which have a lower intrinsic level of evidence compared with prospective studies [30]. Nonetheless, in our opinion this type of research design is a valuable tool to evaluate the impact on developmental profile of early treatment available in community settings to all the children with ASD, in services not supported by research funds. Despite limitations, it provide useful informations may be gathered from study results to direct subsequent prospective studies.“
16.I also think another scale needed to be administered instead of diagnostic scales such as ADOS2 to determine the impacts of the interventions. Some scales such as ATEC ( Autism Treatment Evaluation Checklist). A major obstacle in autism research has been the lack of a valid means of measuring the effectiveness of various treatments. And the application of the diagnostic scales is not recommended for the validity of the intervention approaches. Over the years, researchers have published hundreds of studies attempting to evaluate different biomedical and psycho-educational interventions intended to benefit children with autism. Much of this research produced inconclusive or, worse, misleading results because there are no useful tests or scales designed to measure treatment effectiveness. Lacking such a scale, researchers resorted to using scales such as ADOS ( as it was used in this study) or other scales such as the Childhood Autism Rating Scale (CARS), the Gilliam Autism Rating Scale (GARS), or the Autism Behavior Checklist (ABC), all of which were designed to diagnose autism- to tell whether or not a child has ASD--and not to measure treatment effectiveness.
We agree with the view expressed by the Reviewer, the application of the diagnostic scales is not recommended for the validity of the intervention approaches. As a retrospective chart review by definition as a retrospective study by definition, it is not possible to modify the tests used in the diagnostic phase. Only a prospective open trial design would have allowed pre-/post-testing with a pre-determined set of tests, not the retrospective design of the present study.
The first aim of our study is to verify the impact on developmental profile of different types of early intervention programs delivered by local service; therefore, we choose as comparing measure the Griffiths Mental Developmental Scales-Extended Revised version. Researchers to assess developmental skills often use this useful toll (e.g. Muglia P, Filosi M, Da Ros L, et al. The Italian autism network (ITAN): a resource for molecular genetics and biomarker investigations. BMC Psychiatry. 2018;18(1):369. Published 2018 Nov 21. doi:10.1186/s12888-018-1937-y; Li HH, Wang CX, Feng JY, Wang B, Li CL, Jia FY. A Developmental Profile of Children With Autism Spectrum Disorder in China Using the Griffiths Mental Development Scales. Front Psychol. 2020;11:570923. Published 2020 Nov 9. doi:10.3389/fpsyg.2020.570923). Anyway, the intrinsic limitation due to study design is now specified in the “study limitations” paragraph of our revised Discussion (page 8).
I agree that this is very important to understand the impacts of different intervention approaches, but I also think such studies are very difficult to administer.
I also think that considering the Denver model as a different approach from the Behavioral model is not a very strong premise. The Early Start Denver Model (ESDM) is a behavioral therapy for children with autism between the ages of 12-48 months. This approach is based on the methods of applied behavior analysis (ABA). I also think that the other unclear approach in your study, AUT, might have the ABA components in its intervention. This is a very strong justification for having no statistically significant difference between the three groups in your study. I also think that comparing ESDM with the other nonbehavioral models, such as the Hanen approach, might yield different results. Hanen programs reflect a family-centered model of intervention based on a social-interactionist perspective of language development. The Hanen approach is based on the principles that parents should be involved in their child's intervention process and that communication should be facilitated in naturalistic contexts.
Parents and therapists use play to build positive and fun relationships. Through play and joint activities, the child is encouraged to boost language, social and cognitive skills.
You could also compare clinical-based and home-based approaches to early intervention to understand the possible differences between different methods with different conceptual bases.
In sum, I think that your study used diverse terminology (ESDM, TAU, and EIBI); in reality, they were possible all the same, and the results support this justification. Using different words to express similar meanings can lead to difficulties understanding and implementing the interventions in detail. Particularly no data has been presented at all regarding TAU, and children who received this approach also had no statistically significant difference from the other two groups. This is very crucial to know what was the contents of this approach that was this effective. In particular, you were not indicated which approaches were implemented directly or indirectly by parents or caregivers, who are non-experts, and which approach was pure clinical and presented by the clinicians. This could be an even more significant obstacle to generalizing the present findings. As the Early intervention services provider or researcher, you need to classify and define interventions using standard, clearly, widely understood terms (Behavioral, cognitive, communication, or parental or clinicians based). In the case of interventions that do not belong to an existing classification (in your case, TAU), it will be necessary to introduce the principles and specific implementation methods in more detail.
We agree with the view expressed by the Reviewer that is very important to understand the impacts of different intervention approaches, and even more to verify the efficacy of those different approaches when they are released by community-based settings.
We choose a retrospective chart review to better verify the effectiveness of these treatments in real life, as they are not built into universitary setting and performed only by ultra-specialized experts. Therefore, we are not able to compare ESDM or EIBI intervention with the Hanen approach, because of in our territory there are no qualified therapists who practice this intervention.
About TAU, in lines 72-78 we provide a literature description about public treatment, and below we have added more details about treatment-as-usual group in our study. Moreover, we provide to include this in the limitations of our study.
Round 2
Reviewer 2 Report
This is an updated version of the previously submitted manuscript. The authors have done their best to answer the comments and resolve the issues I made. Hence, this paper still has severe methodological problems, and I think that some of the presented justifications are not at a satisfactory level. For instance, I can mention the given definition for "treatment as usual." The most crucial issue I made was the closeness of the two compared approaches and the unclarity of the third approach (TAU). This critical issue has not been addressed in the limitations. Lack of accessibility to procedures with fundamental differences (different approaches) is an important issue. The compared methods need to be presented and defined for the reader to understand their overlap and differences. This part of the study will also clarify to the reader why these approaches compared with each other and how different or similar they were.
Author Response
Troina - August 30, 2022
Dear Reviewer,
Thank you for your kind consideration of our manuscript entitled “Impact of different kinds of early interventions on develop-mental profile in toddlers with autism spectrum disorder”. We have now revised the manuscript responding to all your comments. Your comments have been addressed as detailed below:
REVIEWER
This is an updated version of the previously submitted manuscript. The authors have done their best to answer the comments and resolve the issues I made. Hence, this paper still has severe methodological problems, and I think that some of the presented justifications are not at a satisfactory level. For instance, I can mention the given definition for "treatment as usual." The most crucial issue I made was the closeness of the two compared approaches and the unclarity of the third approach (TAU). This critical issue has not been addressed in the limitations. Lack of accessibility to procedures with fundamental differences (different approaches) is an important issue.
The compared methods need to be presented and defined for the reader to understand their overlap and differences. This part of the study will also clarify to the reader why these approaches compared with each other and how different or similar they were.
We would like to thank the Reviewer for his interest in our results and for helping us to improve our manuscript. As you suggested, we clearly underlined the critical issue within the limits of the proximity of the two treatments used and the difficulty of defining the TAU. Furthermore, the three treatments analyzed were better defined and the differences between eibi and esdm were highlighted (please see lines 85-107 and 289-301).
Sincerely,
Luigi Vetri